# Solid-Phase Microextraction/Gas Chromatography–Time-of-Flight Mass Spectrometry Approach Combined with Network Pharmacology Analysis to Evaluate the Quality of Agarwood from Different Regions against Anxiety Disorder

**DOI:** 10.3390/molecules29020468

**Published:** 2024-01-17

**Authors:** Yue Pang, Wenjuan Yu, Wenyi Liang, Yu Gao, Fan Yang, Yuanyuan Zhu, Lei Feng, Hongmei Yin, Yumin Liu

**Affiliations:** 1School of Pharmacy, Shanghai Jiao Tong University, Shanghai 200240, China; pangy98@sjtu.edu.cn (Y.P.); lwy0213599@163.com (W.L.); 2Instrumental Analysis Center, Shanghai Jiao Tong University, Shanghai 200240, China; shirlygao@sjtu.edu.cn (Y.G.); yangfan0621@sjtu.edu.cn (F.Y.); zhuyuanyuan@sjtu.edu.cn (Y.Z.); fiona.fenglei@sjtu.edu.cn (L.F.); 3State Key Laboratory of Subtropical Silviculture, Zhejiang A&F University, Hangzhou 311300, China; yuwingjane@gmail.com; 4Hangzhou Institute for Food and Drug Control, Hangzhou 310022, China; fda01@126.com

**Keywords:** agarwood, sesquiterpenoids, quality, SPME/GC-TOFMS, network pharmacology analysis, anxiety disorder

## Abstract

Agarwood (*Aquilaria malaccensis* Lam.) is a resinous material from different geographical locations. The current evaluation of agarwood quality is usually based on its physical properties and chemical compounds, yet only a few studies have linked agarwood quality with its anxiolytic effect, as indicated by characteristic compounds. In this study, using solid-phase microextraction/gas chromatography–time-of-flight mass spectrometry (SPME/GC-TOFMS) and multivariate analysis, we found 116 significantly different compounds in agarwood samples from four locations in Southeast Asia with regard to their quality. Brunei and Nha Trang agarwood had abundant sesquiterpenoids, exhibiting notable pharmacological efficacy in relieving anxiety. Malaysian and Irian agarwood had abundant alcohols and aldehydes, qualifying them as high-quality spices. Compound–target–disease network and pathway enrichment analysis were further employed to predict 79 gene targets and 20 pathways associated with the anxiolytic effects based on the 62 sesquiterpenoids. The correlated relationships among the sesquiterpenoids and targets suggest that agarwood treats anxiety via multiple compounds acting on multiple targets. Varying levels of sesquiterpenes across agarwood groups might lead to differences in the anxiolytic effects via signaling pathways, such as neurotransmitter- and hormone-regulated pathways. Our study originally evaluates agarwood quality and its anxiolytic effect by linking the characteristic compounds to potential gene targets and pathways.

## 1. Introduction

In recent years, an increasing number of people have begun to suffer from anxiety disorder. Some researchers confirmed that agarwood (*Aquilaria malaccensis* Lam.) has great potential to relieve fatigue, aid with sleep, calm the mind, reduce anxiety, and relieve depression and other negative emotions [1,2,3,4]. Agarwood mainly grows in Southeast Asia (Laos, Cambodia, Vietnam, Thailand, Indonesia, and Malaysia) and South China (chiefly in the Hainan and Guangdong provinces). Historically, agarwood has been widely used for incense, perfumes, and traditional medicine around the world. Agarwood from different regions is traded at different prices depending on its quality. Traditionally, agarwood quality has been linked with its physical appearance, such as color, density, and scent [5], yet no standard method is available for this assessment.

Agarwood contains alcohols, aldehydes, ketones, chromones, and several types of sesquiterpenes. Sesquiterpenes are significantly bioactive compounds in agarwood. Sesquiterpenes are primarily categorized into agarofuran, agarospirane, cadinane, eremophilane, eudesmane, guaiene, and prezizane types [6,7,8]. More objectively, numerous volatile and semi-volatile compounds may be the characteristic constituents with key physiological benefits in agarwood [2]. Changes in sesquiterpenoids and/or other compounds in agarwood may lead to the alteration of its biological activity and pharmacology [9,10,11]. Since the formation of agarwood is associated with the wounds and fungal infections of the *Aquilaria* trees, resin production and the quality of agarwood are largely determined by the species, the growing circumstances, state of agarwood formation, and even age [12]. Recently, some researchers attempted to grade agarwood based on its metabolites, including sesquiterpenoids and phenylethyl chromone derivatives. Shaari et al. applied 1H-NMR-based metabolomics to grade the quality of agarwood samples on the basis of their chemical constituents [13]. Suresh et al. carried out a study on different grades of agarwood oil using a GC-MS approach [14]. Yet, these studies did not link the anxiolytic quality of agarwood with its geography, which greatly impacts its metabolite profile [15,16]. More importantly, minimal research has been carried out to characterize the correlation between the volatile and semi-volatile compounds of agarwood from different regions and its anxiolytic effect.

With the rapid development of bioinformatics, a network pharmacology approach has been widely used to explore drug interactions with multiple targets [17,18]. Based on public databases and published biological data, visual compound–therapeutic target–disease networks can be constructed. The constructed network models can explore the mechanisms of the therapeutic effects of traditional medicines. A network approach helped study the possible interactions among 11 characteristic compounds from agarwood and anxiety- and depression-related diseases; 30 therapeutic gene targets suggested that agarwood essential oil can regulate neural activity [19]. Nevertheless, it remains unclear whether and how agarwood from different regions exerts different effects on the nervous system. Thus, network pharmacological analysis provides a great opportunity to evaluate and predict distinctive pharmacological effects caused by different agarwood compounds from various regions.

To address these knowledge gaps, this work aims to investigate the differences in volatile and semi-volatile compounds among the agarwood from Brunei, Nha Trang, Malaysia, and Irian using SPME/GC-TOFMS and multivariate analysis. Then, the work aims to evaluate the anxiolytic effect of agarwood through the network pharmacological analysis of some interesting sesquiterpenoids. Finally, the work aims to explore the potential relationships between agarwood quality and its anxiolytic effect by linking the characteristic compounds to potential gene targets and pathways.

## 2. Results

### 2.1. Identification of Volatile and Semi-Volatile Compounds of Agarwood

A total of 372 volatile and semi-volatile compounds were identified (Appendix A). These compounds belong to 9 chemical classes: 8 furans, 29 alcohols, 20 aldehydes, 33 ketones, 18 acids, 16 esters, 30 monoterpenes, 118 sesquiterpenoids, and 100 other compounds. The main sesquiterpenes contain 4 agarofurans, 2 agarospiranes, 2 prezizaane, 14 guaianes, 16 eudesmanes, 9 eremophilanes, 10 bisabolanes, 9 cadinanes, 7 aromadendranes, 4 humulanes, 6 cedranes, 4 elemanes, 3 acoranes, 3 longifolanes, 3 germacrane, 3 caryophyllanes, 1 aristolanes, 1 copanes, 1 maalianes, 1 thujopsanes, 1 lauranes and 14 others.

### 2.2. Relationships among Agarwood Groups and Compounds as Indicated by PLS-DA Biplot of Scores and Loadings

The GC-TOFMS total ion current (TIC) profiles of agarwood across regions were somewhat similar, but the intensity peaks were different (Appendix A). The distance among the four agarwood groups in the PCA and PLS-DA models (Appendix A) showed that the agarwood samples from the same region had similar volatile profiles, while the overall agarwood compounds differed among different regions. In the PLS-DA model, the R^2^Y and Q^2^ parameters were used to assess its fitness and prediction ability, respectively. The variances explained by the components (R^2^X and R^2^Y) of the models were 38.3% and 95.9%, respectively, and the prediction accuracy (Q^2^Y) was 84.6% through typical sevenfold cross-validation. R^2^Y and Q^2^ were close to one. These results indicate that the model was relatively stable and had a good prediction ability. In Figure 1, the PLS-DA biplot of scores and loadings further show correlations among the 372 compounds and agarwood groups. The samples from Brunei and Nha Trang are far from each other, while those from Malaysia and Irian are close to each other. Many critical sesquiterpenoids in agarwood were strongly correlated with the Brunei group, which is clustered in the upper left quadrant of the biplot. These sesquiterpenoids include nootkatone, α-agarofuran, β-dihydroagarofuran, agarospirol, rosifoliol, γ-eudesmol, eudesma-4(15),7-dien-1β-ol, etc. Several sesquiterpenes, aldehydes, and acids are close to the Irian group located in the center, such as α-selinene, valerenal, (E)-isovalencenal, α-ylangene, isocaryophyllen, γ-amorphene, hexanal, p-tolualdehyde, furfural, heptanoic acid, octanoic acid, etc. Near the Malaysia group, the benzenes, alcohols, and monoterpenoids, including 2-phenylisopropanol, veratraldehyde, benzyl alcohol, morillol, 1-decanol, 1-octanol, and camphor, are clustered in the upper right quadrant. The Nha Trang group is correlated to the majority of the compounds clustered in the lower right quadrant. These compounds covered short-chain acids, aldehydes, and sesquiterpenoids, such as propanoic acid, heptanal, 4-epi-cis-dihydroagarofuran, epiglobulol, α-bisabolol, α-eudesmol, etc.

### 2.3. Differential Compounds across Four Groups of Agarwood

A total of 116 compounds across four groups of agarwood had both VIP > 1.0 in the PLS-DA model and *p* < 0.05 in Duncan’s multiple comparisons post hoc tests. The relative intensities of the 116 differential compounds across four agarwood groups are shown in a heatmap (Figure 2). The cell in the heatmap represents the intensity value of a compound averaged across all the agarwood samples in a type of agarwood. Among these 116 compounds, 52, 19, 29, and 16 compounds were most abundant in the Brunei, Nha Trang, Malaysia, and Irian groups, respectively. Since the sesquiterpenes comprise a majority of pharmacologically effective compounds of agarwood, we focused on 62 sesquiterpenoids out of the 116 differential compounds (Appendix A). These sesquiterpenes can be mainly categorized into seventeen types with distinct physicochemical properties, including agarofurans, agarospiranes, guaianes, eudesmanes, eremophilanes, acoranes, humulanes, aromadendranes, elemanes, cedranes, cadinanes, bisabolanes, copanes, caryophyllanes, thujopsanes, longifolanes, and germacranes. In Figure 2, 40 out of 62 sesquiterpenoids had the highest intensities in Bruneian agarwood among the four groups, such as α-agarofuran, β-dihydroagarofuran, agarospirol, rosifoliol, and α-guaiene. Isocalamenediol, five out of eight monoterpenoids, five out of seven alcohols, and three out of eight aldehydes were more abundant in the Malaysia group than those in the other three groups. Nha Trang agarwood had the highest intensities of several important volatile compounds, such as β-caryophyllene, α-eudesmol, epiglobulol, α-bisabol, and propanoic acid. Irian agarwood had the highest intensities of some sesquiterpenoids, three out of eight aldehydes, and two out of six furans, including α-selinene, γ-amorphene, isocaryophyllene, (E)-isovalencenal, α-ylangene, isogermacrene D, clovene, furfural, benzaldehyde, and p-tolualdehyde.

### 2.4. Network Pharmacology Analysis

#### 2.4.1. Compound–Target–Disease Network Construction

To explore the anxiolytic effects of four types of agarwood, we used network pharmacological analysis to evaluate and predict the targets of the differentially expressed sesquiterpenoids in agarwood. First, 254 pharmacological target genes were identified as related to the 62 differential sesquiterpenoids (Appendix A) across four agarwood groups via the SwissTargetPrediction database. Next, 1942 target genes linked to anxiety disorders were mined from these online databases, GeneCards, OMIM, and Drugbank. A total of 79 genes were found in both gene groups (Appendix A), suggesting that these 79 genes may be the potential targets that exert agarwood’s anxiolytic effects. To further understand the drug–target interaction mechanism of agarwood, we built a network to visualize the compound–target–disease correlations. As shown in Figure 3, the network contained 143 nodes and 775 edges. The averaged network connectivity was 11.825, and the averaged network density was 0.076. One blue node, one purple node, sixty-two orange nodes, and seventy-nine Kelly nodes represent agarwood, anxiety disorder, sesquiterpenoids, and the overlapping targets between anxiety disorder and agarwood, respectively. Seven hundred and seventy-five black connections (edges) indicate correlated relationships among the sesquiterpenoids and targets. The core genes with degrees ≥ 20 were screened out, including CYP19A1, AR, ESR1, PPARA, SLC6A4, SHBG, UGT2B7, ACHE, SLC6A2, BCHE, ESR2, CHRM2, and CYP2C19 (Figure 3). Likewise, 12 important compounds with degrees ≥ 20 were considered to be important agarwood constituents that can potentially improve anxiety disorders (Table 1). These representative compounds exhibited significant differences among the agarwood groups (Figure 4). For example, nootkatone had the highest connection degree of 31; α-bisabolol and agarospirol also had high degrees of 24 and 22, respectively (Table 1). These findings suggest that some of the important sesquiterpenoids in agarwood could exert anxiolytic effect via multiple targets.

#### 2.4.2. Pathway Enrichment Analysis

To further identify the potential pathways involved in the anxiolytic effect of agarwood, KEGG pathway enrichment analysis was performed on the Metascape database to study the abovementioned 79 target genes possibly related to the agarwood sesquiterpenes. A total of 20 significantly enriched pathways (*p* < 0.001) involved in the anxiety disorder effects of agarwood were identified and are presented in Figure 5. The 79 target genes were linked to a variety of signaling pathways, including neuroactive ligand–receptor interaction, pathways in cancer, steroid hormone biosynthesis, chemical carcinogenesis–receptor activation, serotonergic synapse, calcium signaling pathway, type II diabetes mellitus, arginine and proline metabolism, morphine addiction, pathways of neurodegeneration-multiple disease, insulin resistance, cholinergic synapse, platelet activation, lipid and atherosclerosis, retrograde endocannabinoid signaling, synaptic vesicle cycle, circadian rhythm, aldosterone-regulated sodium reabsorption, inflammatory mediator regulation of TRP channels, and bile secretion. Among these pathways, the neuroactive ligand-receptor interaction, serotonergic synapse, calcium signaling pathways might be closely related to the occurrence and treatment of anxiety disorder.

## 3. Discussion

Despite the sample chromatograms having similar overall chemical profiles, we discovered significant differences in the sesquiterpenoids among the agarwood from four origins using the SPME-GC/TOFMS method and multivariate statistics. The pharmacological activities of the agarwood samples of different geographical origins are related to the characteristic sesquiterpenes, which are affected by many factors, including temperature, light, wind, moisture, soil, and the gnawing of ants [20,21]. Other studies have shown that some sesquiterpenoids can exert sedative, anxiolytic, and antidepressant effects. The sesquiterpenoids also contribute to the formation of the unique aroma of agarwood [1,2,3,10]. Sixty-two sesquiterpenoids obtained from four types of agarwood are probably important “gateways” affecting the pharmacological properties of agarwood.

### 3.1. Unique Sesquiterpenoids with Pharmacological Activities in Four Agarwood Groups

The multivariate results showed that four agarwood groups tended to cluster by origin based on the variations in volatile composition (Figure 1 and Figure 2). Bruneian agarwood had the highest peak intensities of most of the sesquiterpenoids among the investigated agarwood samples (Figure 2). According to the predicted therapeutic targets using network pharmacology, agarospirol, nootkatone, rosifoliol, proximadiol, eudesma-4(15),7-dien-1β-ol, epi-γ-eudesmol, γ-eudesmol, and torreyol are potentially bioactive compounds contributing to the sedative and anxiolytic effects of agarwood, and they were all significantly more abundant in Bruneian agarwood than they were in the other three types of agarwood (Figure 4 and Appendix A). Agarospirol, an agarospirane-type sesquiterpenoid, is a characteristic compound of agarwood with a distinctive woody, peppery, and spicy odor [16]. Playing a key role in agarwood’s pharmacological effect, agarospirol has been shown to have a calming effect on the nerve centers of mice, as well as anti-inflammatory and immunomodulatory effects [22,23]. Rosifoliol, an eudesmane-type sesquiterpenoid, combined with agarospirol has anxiolytic and antidepressant effects by acting on the 5-HT_1A_ receptor associated with mood alterations [24]. Nootkatone, an eudesmane-type sesquiterpene, has been reported to have a strong grapefruit aroma and various neuroprotective effects [25,26]. Yan et al. discovered that Nootkatone could improve anxiety- and depression-like behavior by inhibiting oxidative stress and simultaneously enhancing hippocampal neurogenesis in mouse brain tissues [27]. Besides the important sesquiterpenes identified in the network analysis, others are also bioactive. β-dihydroagarofuran, an agarofuran-type sesquiterpenoid, had the highest intensity in the Bruneian agarwood group. Ishola et al. observed that β-dihydroagarofuran could produce antidepressant and anxiolytic-like functions by enhancing monoaminergic signaling [28]. In addition, α-guaiene, a Guaiacane-type sesquiterpene, was more abundant in the Bruneian agarwood samples and showed remarkable inhibitory effects against cyclooxygenase, 5-lipoxygenase, and acetyl cholinesterase enzymes in the treatment of inflammatory-related ailments and cognitive disorders [29]. Given the presence of these sesquiterpenoids playing important roles in the anxiolytic activity of agarwood, Bruneian agarwood seems to be a valuable herbal medicine with a great potential to treat anxiety.

In the biplot of the PLS-DA model, the Nha Trang group is close to α-eudesmol, α-bisabolol, β-vatirenene, β-caryophyllene, and gossonorol (Figure 1). These compounds were more abundant in the Nha Trang group than in the three other groups (Appendix A). α-Eudesmol is a sesquiterpenoid alkene alcohol, which may be effective against the development of pain and neurogenic inflammation [30]. Asakura et al. demonstrated that α-eudesmol might inhibit neural hyperactivity by blocking the presynaptic ω-agatoxin IVA-sensitive (P/Q-type) Ca^2+^ channel in neurons [31,32]. α-Bisabolol also has certain anti-inflammatory, antinociceptive, and anxiolytic effects [33,34]. Tabari et al. tested the effect of α-bisabolol on anxiolytic-like behaviors among mice. The open-field experiment showed that α-bisabolol produced a sedative effect on mice at high doses and an anxiolytic effect at low doses via GABAergic transmission [35]. β-Caryophyllene, which belongs to the bicyclic sesquiterpenes with anti-inflammatory, antioxidant, and anxiolytic activities, have been reported to play an important neuroprotective role in neurological diseases, including anxiety, convulsion, depression, and Alzheimer’s disease [36]. However, the levels of representative sesquiterpenes of Nha Trang agarwood (e.g., agarofurans, agarospiranes, prezizaane, and guaianes) were lower than those in Bruneian agarwood (Figure 2). Our findings thus suggest that although the above sesquiterpenoids can play an important role in regulating the central nervous system, the quality of Nha Trang agarwood as indicated by the characteristic sesquiterpenes might be a little inferior to that of Bruneian agarwood.

In the biplot, the Malaysia group is close to most of the alkanes and alcohols (Figure 1). The heatmap shows that the Malaysian agarwood samples are indeed characterized by high levels of some alcohols (Appendix A). Alcohols typically contribute to various aromas. For example, 1-decanol, 1-octanol, benzyl alcohol, and morillol have floral rose scents, a citrus aroma, a fruity odor, and a mushroom-like perfume similar to plant freshness, respectively [37,38]. Among these alcohols, a rose aroma can lower a person’s blood pressure and heart rate and calm people down [39]. In addition, Malaysian agarwood also contained a high level of camphor, a monoterpenoid compound with a distinct camphor scent, which can refresh the brain [40]. But the contents of most sesquiterpenoids in the Malaysian agarwood group were relatively low among the four agarwood groups (Figure 2). We thus suggest that Malaysian agarwood might have a stronger aroma. As a crucial spice source, Malaysian agarwood oil is highly valued in Southeast Asian countries [14,16,41]. 

Furfural, benzaldehyde, and p-tolualdehyde were more abundant in the Irian group than those in the other groups (Appendix A). Benzaldehyde and furfural are important contributors to the overall aroma of agarwood essential oil, with a nutty aroma similar to that of almonds [16]. p-Tolualdehyde has a pleasantly cherry and fruity taste [42]. In Figure 2, the heatmap further indicates that the contents of the sesquiterpenoids, such as isocaryophyllen and γ-amorphene, were higher in Irian agarwood than those from the other three regions. However, these compounds are not the main ingredients responsible for the anxiolytic effect of agarwood. The Malaysia and Irian groups had many aromatic compounds and similar overall volatile compositions (Figure 1), suggesting that they are more suitable as spices than medicines.

### 3.2. Potential Therapeutic Targets for Anxiety Treatment as Indicated by Compound–Target–Disease Network

Network pharmacology contributes to the exploration of the relationships among herbs, diseases, and molecular targets. In this study, we created a compound–target–disease network by establishing connections between 62 differential sesquiterpenoids from agarwood of four regions and the potential therapeutic targets for treating anxiety disorders. The network displayed 79 overlapping genes that might serve as molecular targets via which agarwood helps with anxiety (Figure 3). The correlated relationships between the sesquiterpenoids and targets suggest that agarwood potentially treats anxiety disorder, with multiple compounds acting on multiple targets. Based on 79 targets genes, we further highlighted 20 gene pathways that might be affected by the agarwood sesquiterpenes. As shown in Figure 5, the neurotransmitter-related pathways are neuroactive ligand–receptor interaction, serotonergic synapse, cholinergic synapse, and synaptic vesicle cycle. The hormone signaling pathways include steroid hormone biosynthesis and aldosterone-regulated sodium reabsorption. The calcium signaling pathway and pathways of neurodegeneration-multiple diseases are related to the signal transduction pathway and neurological disorders, respectively. These pathways involve neurotransmitter, hormone, and calcium signaling, which are all key players in treating anxiety. The varied sesquiterpenes across the agarwood groups are thus expected to differentially affect the gene targets and their pathways, leading to different anxiety treatment effects among the groups.

In Figure 3, the core genes with a degree value ≥ 20 included CYP19A1, AR, SHBG, UGT2B7, ESR1, and ESR2, which are all relevant to the regulation of estrogens [43,44,45,46,47,48]. It is well documented that estrogen might exert profound effects on anxiety and anxiety-like behaviors in humans and rodents [49]. In behavioral trials, such as light- and dark-box and open-field tests in animals, it was shown that the activation of the estrogen receptors ESR1 and ESR2 is closely related to increased anxiety in rodents. When the estrogen receptors were downregulated, the anxiety-like behaviors were reduced [50,51]. Estrogen can also regulate several serotonergic systems, including facilitating serotonin neurotransmission and controlling the expression of tryptophan hydroxylase (TPH), the rate-limiting enzyme for serotonin synthesis [52]. Serotonin (5-HT) plays an important role in physiological states, such as cognitive function, mood disorders, and anxiety disorders [53,54]. An important physiological role of BCHE gene (degree > 20) is to reduce the background levels of ghrelin in the brain to control anxiety and depression. BCHE encodes butyrylcholinesterase, which breaks down acetylcholine and ghreline. Previous studies have shown that ghrelin enhances fear and anxiety after acute and chronic stress exposure [55]. In summary, our network analysis indicates that these core genes could serve as potential therapeutic targets for anxiety treatment in the future.

The anxiolytic effect of sesquiterpenoids may be related to the neurotransmitter-related pathways involved in neuroactive ligand–receptor interaction and serotonergic synapse signaling (Figure 5). One monoamine neurotransmitter is 5-HT. Its serum level is associated with anxiolytic and antidepressant effects of agarwood, possibly via multiple neuroactive pathways, including serotonergic synapse and neuroactive ligand–receptor interaction [56]. Wang et al. also found that β-caryophyllene (one out of sixty-two sesquiterpenoids in agarwood) might activate the serotonergic synapse signaling pathway to increase the activity or affinity of the 5-HT, thus alleviating anxiety [57]. In another study on the anxiolytic and antidepressant effects, a mixture of rosifoliol and agaraspirol with different doses was reported to have some influence on the 5-HT_1A_ receptor in the serotonergic system [24]. Consistent with these reports, the genes serotonin (SLC6A4) and norepinephrine transporter (SLC6A2) found in our study were linked to the vulnerability to mental diseases and antidepressant effect [58,59]. In addition, the calcium signaling pathway was also located at the top of the bubble charts in the KEGG pathway enrichment analyses (Figure 5). α-Eudesmol (degree 20) is a type of calcium channel blocker and plays a key role in neuronal functions by inhibiting exocytotic glutamate release and facilitating brain edema formation [30,31,32]. Overall, the sesquiterpenoid differences in the agarwood from four regions may act differently toward the abovementioned anxiety targets via multiple signaling pathways, providing directions for future experimental confirmation.

## 4. Materials and Methods

### 4.1. Sample Preparation

All the natural agarwood samples used in this study were obtained from a reliable agarwood supplier (Shanghai Qinan Co., Ltd., Shanghai, China). Malaysian agarwood was collected in Kelantan, Malaysia; Nha Trang agarwood was collected in Nha Trang, Vietnam; Bruneian agarwood was collected in Temburong, Brunei; and Irian agarwood was collected in Jayapura, Indonesia. Each group contained 10 representative samples from the same region. The samples were inspected for authenticity by an experienced collector and trader, and further identified by Prof. Hongmei Yin (Hangzhou Institute of Food and Drug Control, Hangzhou, China). The specimens were deposited at the Instrumental Analysis Center of Shanghai Jiaotong University (Appendix A).

All the samples were crushed with a grinding machine (Shanghai Jingxin Industrial Development Co., Ltd., Shanghai, China). A total of 10 mg of each agarwood sample powder was placed in a 20 mL headspace vial and sealed for the following analysis.

### 4.2. GC-TOFMS Analysis

A CTC autosampler (CTC Analytics AG, Zwingen, Switzerland) was employed for the SPME of the samples. After shaking and incubating at 70 °C for 10 min, the volatile and semi-volatile compounds of agarwood were extracted with a 2 cm 50/30 divinylbenzene/carboxen/polydimethylsiloxane (DVB/CAR/PDMS) fiber (Supelco, Bellefonte, PA, USA) at 70 °C for 40 min, followed by desorption in a GC injector at 250 °C for 5 min [60]. An Agilent 7890 gas chromatograph (Agilent Technologies, Palo Alto, CA, USA) coupled with a Pegasus BT time-of-flight mass spectrometer (Leco Corp., Santa Clara, CA, USA) was used to perform the aroma analysis. The samples were separated on a DB-WAX column (30 m × 250 μm × 0.25 μm; Agilent J&W Scientific, Folsom, CA, USA). Helium (99.999%) was used as a carrier gas at a rate of 1.0 mL/min. The GC oven was initially held at 40 °C for 5 min, and then ramped up to 220 °C at 5 °C/min, and finally to 250 °C at 20 °C/min for 7.5 min. Transfer line and ion source were set at 260 °C and 230 °C, respectively. The TOFMS data were acquired with electron impact ionization (70 eV) in full scan mode (*m*/*z* 29–450). Each scan was set at a rate of 10 spectra per second. A mixture of C7-C40 alkanes (Sigma Aldrich Trading Co., Ltd., Shanghai, China) was analyzed using the same GC-MS method and used to calculate the retention index (RI) of the peaks.

### 4.3. Compound Identification and Quantification

All the GC-TOFMS data files were converted to CDF format, followed by automatic peak detection and mass spectrum deconvolution using ChromaTOF software (version 5.50, Leco Corp., CA, USA). The peaks with a signal-to-noise (*S*/*N*) ratio < 30 were rejected. The retention index of each compound was also calculated based on the retention time of C7–C40 alkanes using ChromaTOF software. The compounds were identified via a comparison of their mass spectra and retention indices with those in the National Institute of Standards and Technology (NIST) mass spectral library (version 20). A mass spectrum similarity threshold of 700/1000 was used for this purpose. LECO ChromaTOF software was further used to process the original data into a mathematical matrix via steps of peak area integration, baseline calibration, and filtering. Using the area normalization method, the area of each peak was divided by the total area of all peaks to obtain the relative peak intensity.

### 4.4. Statistical Analysis

The final mathematical matrix containing sample information, peak retention time, and normalized peak intensity was imported into SIMCA-P 14.0 (Umetrics, Umeå, Sweden) for multivariate statistical analyses. These statistical models were used to assess the variations in the compounds identified across the agarwood samples. The differences among four agarwood groups were visualized via principal component analysis (PCA) and partial least squares discriminant analysis (PLS-DA). In the latter model, R^2^X was the cumulative modeled variation in X matrix (the relative intensities of the GC/TOFMS peaks), R^2^Y was the cumulative modeled variation in Y matrix (four different agarwood groups), and Q^2^Y was the cumulative predicted variation in Y, according to cross-validation. A biplot of scores and loadings further visually summarizes correlations among variables in and between the X and Y matrices.

The potential different compounds were first screened according to variable importance in the projection (VIP) in the PLS-DA. When the VIP values of the variables are greater than 1.0, they are considered possibly responsible for group differentiation. Univariate statistical analyses were performed to further screen the differential compounds using SPSS software (IBM SPSS Statistics 24.0). One-way analysis of variance (ANOVA) with Duncan’s multiple comparison method was applied to assess whether compounds significantly (*p* < 0.05) differed among the four groups. The compounds (VIP > 1.0 and *p* < 0.05) were considered as differential ones, and their relative intensities are shown in a heatmap generated using the heatmap illustrator Hemi 3.0.

### 4.5. Network Analysis

A compound–target–disease network was established by linking the differential sesquiterpenoids with their potential targets and associated diseases. Chemical information of the above differential sesquiterpenoids (structure and specification name) was obtained from the Chemical Book (http://www.chemicalbook.com, accessed on 16 February 2022) and NCBI PubChem database (PubChem (nih.gov), accessed on 16 February 2022), and then imported into the SwissTargetPrediction database (SwissTargetPrediction) [61] for further prediction of their potential molecular targets. Meanwhile, the anxiety disorder-associated targets were comprehensively collected from three databases, including the GeneCards (https://www.genecards.org, accessed on 16 March 2022), OMIM (https://www.omim.org, accessed on 16 March 2022) [62], and Drugbank databases (https://go.drugbank.com, accessed on 16 March 2022) [63]. The obtained sesquiterpenoid-related targets were intersected with the anxiety disorder-related targets to construct a Venn diagram of the intersected targets. Based on the interaction of the agarwood, sesquiterpenoids, molecular targets, and anxiety disorder, a complex information network was built and visualized using Cytoscape 3.9.0 software [64]. In the network, the nodes represent the agarwood, sesquiterpenoids, and potential targets, respectively, and the edges represent the compound–target interactions. The network structure is defined by the degrees of connections between the targets and sesquiterpenoids. The higher the degree value, the more important the sesquiterpenoid or target in the network. Important targets or sesquiterpenoids with degrees ≥ 20 were filtrated out for further analysis. Finally, using the Metascape database of multiple targets (https://metascape.org/gp/index.html#/main/step1, accessed on 26 September 2022) [65], KEGG (Kyoto Encyclopedia of Genes and Genomes) signaling pathway enrichment analyses were performed to verify the functional categories of the screened significant targets (*p* < 0.05). The relationship between the KEGG pathways and the genes they encompass was further visualized using the online bioinformatics mapping website (http://www.bioinformatics.com.cn/, accessed on 26 September 2022) [66,67].

## 5. Conclusions

This study showed that the overall volatile composition was similar among the agarwood of four origins, yet the relative intensities of some bioactive compounds were significantly different. Network pharmacology analysis further suggested that varying levels of multiple sesquiterpenoids across the agarwood groups would exert different effects on anxiety through multiple gene targets and their related signaling pathways, including the neurotransmitter-related, hormone-regulated, and signal transduction pathways. Bruneian and Nha Trang agarwood might have good sedative and anxiolytic effects, while Malaysian and Irian agarwood are good spice sources, as indicated by their characteristic volatiles and sesquiterpenoids. This study provides valuable information for the assessment of agarwood quality by linking the characteristic compounds to the potential gene targets and pathways. 

## Figures and Tables

**Figure 1 molecules-29-00468-f001:**
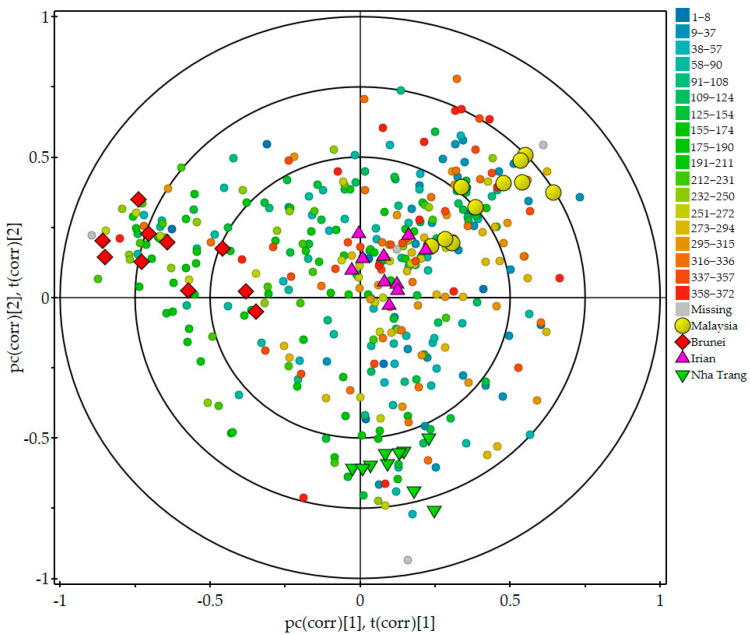
PLS-DA biplot of scores and loadings showing correlations among agarwood samples of different geographical origins and peak intensities of 372 annotated compounds. Each group contains 10 agarwood samples. The detected 372 compounds are colored according to chemical groups (see Appendix A for detailed information on these compounds and their corresponding numbers). Yellow circle, Malaysia group; red diamond, Brunei group; purple triangle, Irian group; green inverted triangle, Nha Trang group.

**Figure 2 molecules-29-00468-f002:**
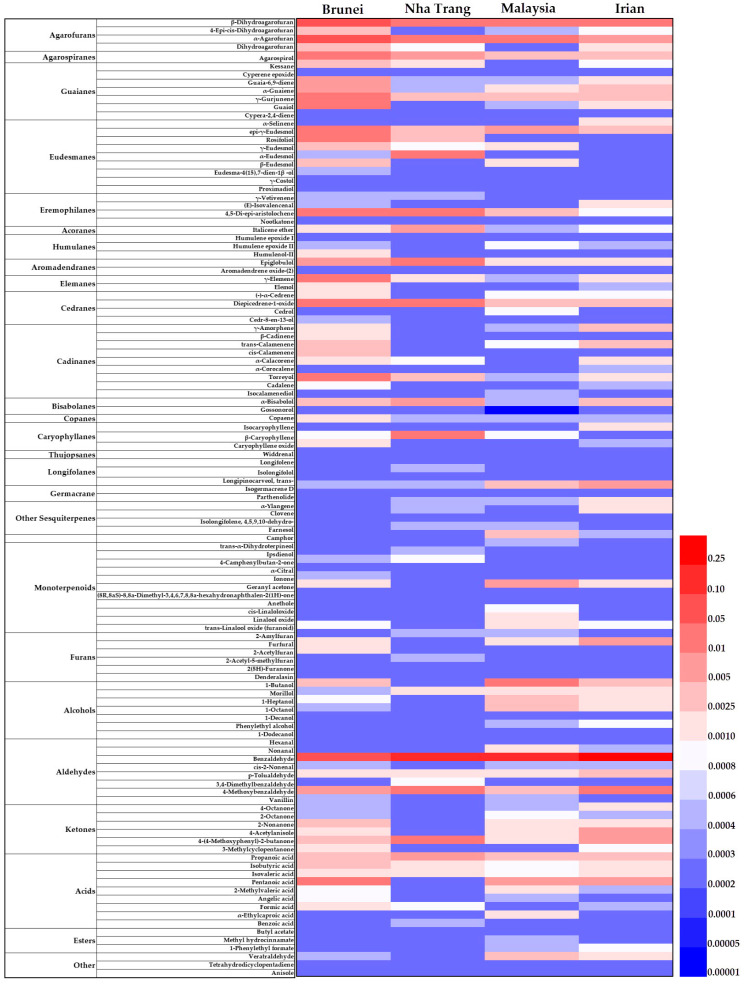
Heatmaps of the relative intensities of 116 differential compounds (PLS-DA VIP > 1.0 and ANOVA *p* < 0.05) among four agarwood groups. The color of each section corresponds to an average intensity value of each compound calculated with the peak area normalization method. The red colors indicate relatively high intensities, while the blue colors indicate relatively low intensities.

**Figure 3 molecules-29-00468-f003:**
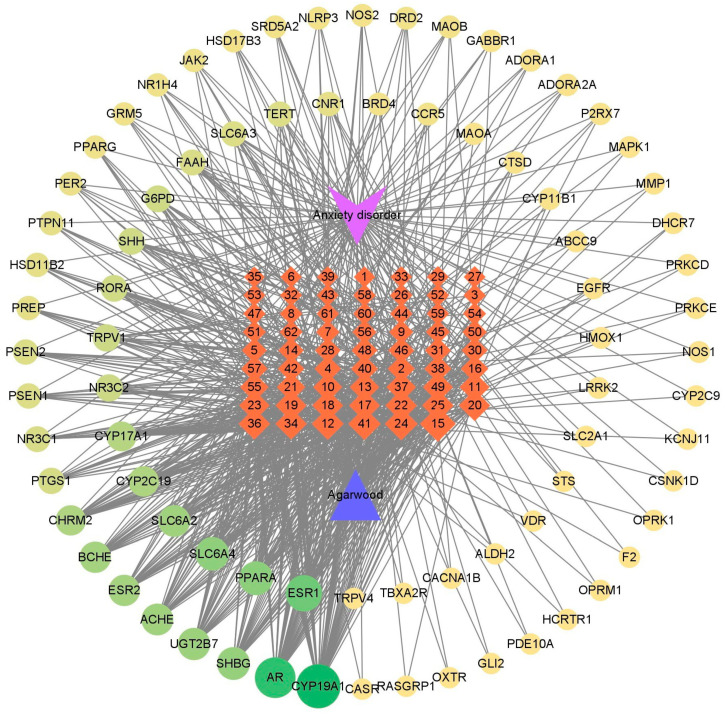
Compound–target–disease network. The purple node represents anxiety disorder; the blue node represents agarwood; 62 orange nodes represent sesquiterpenes with significant differences across agarwood groups (see Appendix A for detailed information on these compounds and their numbers); and 79 green nodes represent overlapping genes. The edges among the nodes indicate their direct interactions. The larger the node, the darker the color, and the more important the node.

**Figure 4 molecules-29-00468-f004:**
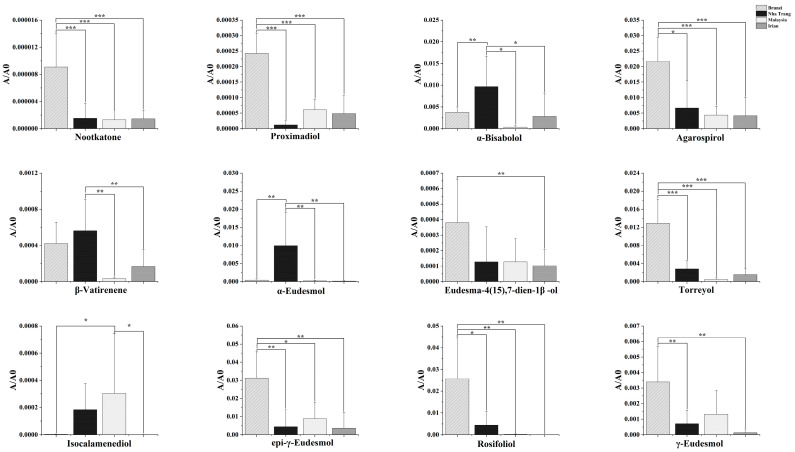
Relative intensities of 12 important sesquiterpenes with degrees ≥ 20 across different agarwood groups. Data are presented as mean ± SD (*n* = 10 for each group). * denotes *p* < 0.05; ** denotes *p* < 0.01; *** denotes *p* < 0.001.

**Figure 5 molecules-29-00468-f005:**
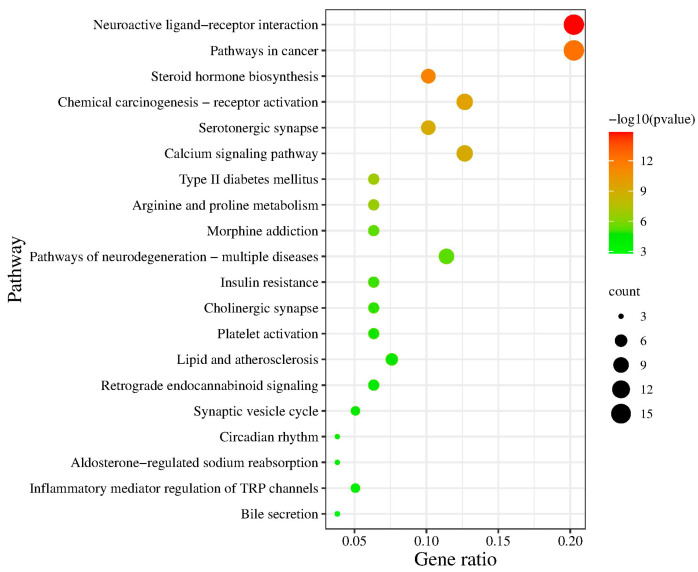
The bubble chart of Kyoto Encyclopedia of Genes and Genomes pathway enrichment analyses. The bubble size reflects the number of target genes in a pathway. Gene ratio is the number of target genes reported in a pathway over 79 (total of screened target genes). The greater the bubble, the higher the gene ratio. The bubble color indicated pathway significance, changing from red (very small *p*) to green (relatively large *p*).

**Table 1 molecules-29-00468-t001:** Twelve sesquiterpenoids with degrees > 20 in the compound–target–disease network.

Name	CAS	Degree
Nootkatone	4674-50-4	31
Proximadiol	4666-84-6	25
α-Bisabolol	515-69-5	24
β-Vatirenene	27840-40-0	23
α-Eudesmol	473-16-5	22
Eudesma-4(15),7-dien-1β-ol	119120-23-9	22
Agarospirol	1460-73-7	22
Torreyol	19435-97-3	22
Isocalamenediol	25330-21-6	22
epi-γ-Eudesmol	117066-77-0	21
Rosifoliol	63891-61-2	21
γ-Eudesmol	1209-71-8	21

## Data Availability

Data are contained within the article and Appendix A. Furthermore, we have preserved all the raw data on our laboratory in Shanghai Jiao Tong University.

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
