# Peer review of "Solid-Phase Microextraction/Gas Chromatography–Time-of-Flight Mass Spectrometry Approach Combined with Network Pharmacology Analysis to Evaluate the Quality of Agarwood from Different Regions against Anxiety Disorder"

_molecules, 2024, doi:10.3390/molecules29020468_

Round 1

Reviewer 1 Report

Comments and Suggestions for Authors

The manuscript entitled “SPME/GC-TOFMS Approach Combined with Network Pharmacology Analysis to Evaluate the Quality of Agarwood from Different Regions Against Anxiety Disorder” identified the differences between the volatiles and semi-volatile compounds in agarwood from different regions. Furthermore, the authors used network pharmacological analysis to evaluate the relationship between the sesquiterpenoids and the anxiolytic effect of agarwood.

In light of the research's reliance on compound identification and the subsequent determination of their relative concentrations for the purpose of conducting network pharmacology analysis, it is imperative to initially validate the accurate identification of these compounds. Regrettably, supplemental information substantiating this validation is presently unavailable.

With respect to the results and discussion, Figure 1 suffers from a lack of visual clarity. Furthermore, we acknowledge the need for more comprehensive explanations of the findings presented in Figures 2, 3, and 5.

A further explanation of the methodology used in the network analysis is necessary.

To my view, the manuscript in its present form is not suitable for publication to Molecules.

Author Response

Thank you very much for taking the time to review this manuscript. Please find the detailed responses below and the corresponding revisions in the re-submitted files.

Reviewer 2 Report

Comments and Suggestions for Authors

Abstract

L14.... has an active effect on the nervous system, yet the pharmaceutical effect... >>> active effect pharmaceutical effect? 

why 4x active? which activity ? CNS??? please thoroughly check the sentence in terms of logic as well!

suggestion:... shows nervous system activity, however the mechanism and mode of actions is not known....? 

L19...differential... > I doubt that this terminology is correct!

L19-20....might provide useful insights into the correlations between the differential compounds and the quality of agarwood against anxiety disorder.>>> the sentence is vague please improve and write a proper CONCLUSION!  may have correlations components and the quality of agarwood preparations...

introduction

L25 (Aquilaria malaccensis)>>> (Aquilaria malaccensis Lam.) please add the scientific name to the mans.

L30-32 have a format problem.. please allign correctly.

L57 .. To address these knowledge gaps, our work>>> To address the knowledge gaps, the present work..

in the results part, the authors should highlight their orginal findings? what was discovered for the first time?

The mans needs major revisions before it can be considered for publication

Author Response

Thank you very much for taking the time to review this manuscript. Please find the detailed responses below and the corresponding revisions in the re-submitted files.

Reviewer 2's comments:

  1. Abstract

Comments 1:  L14.... has an active effect on the nervous system, yet the pharmaceutical effect... >>> active effect pharmaceutical effect? 

why 4x active? which activity ? CNS??? please thoroughly check the sentence in terms of logic as well!

suggestion:... shows nervous system activity, however the mechanism and mode of actions is not known....? 

Response: We agree with the reviewer’s comments. We have corrected the abstract on Line 13. “Agarwood is a resinous material from different geographical locations. Current evaluation of agarwood quality is usually based on its physical properties and chemical compounds, yet few study has linked agarwood quality with its anxiolytic effect as indicated by characteristic compounds. In this study, using solid-phase microextraction / gas chromatography-time of fight mass spectrometry (SPME/GC-TOFMS) and multivariate analysis, we found 116 significantly different compounds in agarwood samples across four locations  in Southeast Asia with regards to their quality. Brunei and Nha Trang agarwood had abundant sesquiterpenoids such as agarospirol, nootkatone, α-eudesmol, exhibiting notable pharmacological efficacy in relieving anxiety. Malaysia and Irian agarwood had abundant alcohols and aldehydes, qualifying them as high-quality spices. A  “compound-target-disease” network and pathway enrichment analysis were further employed to predict 79 gene targets and 20 pathways associated with the anxiolytic effects based on the 62 sesquiterpenoids. Correlated relationships among sesquiterpenoids and targets suggest that  agarwood treats anxiety with multiple compounds acting on multiple targets. Varying levels of sesquiterpenes across agarwood groups might lead to differences in anxiolytic effects via signaling pathways such as neurotransmitter- and hormone-regulated pathways. Our study originally evaluate agarwood quality and its anxiolytic effect by linking characteristic compounds to potential gene targets and pathways. ”

Comments 2: L19...differential... > I doubt that this terminology is correct!

Response: We thank the reviewer for this correction, which we respectfully disagree with. We have cited two papers here to show that “differential” is a commonly-used terminology in metabolomics field to express that some biomarkers have significant differences across groups. Based on reviewer’s comments, we revised the description of abstract on Line 15. We have corrected  “In this study, using solid-phase microextraction / gas chromatography-time of fight mass spectrometry (SPME/GC-TOFMS) and multivariate analysis, we found 116 significantly different compounds in agarwood samples across four locations  in Southeast Asia with regards to their quality. ”

  • Sreekumar,; Poisson, L.M.; Rajendiran, T.M.; Khan, A.P.; Cao, Q.; Yu, J.D.; Laxman, B.; Mehra, R. Lonigro,R.J.; Yong Li, Nyati,M.K.;  Ahsan, A., Kalyana-Sundaram S. ; Han, B.; Cao, X.H.; Byun, J.; Omenn G.S. ; Ghosh, D.; Pennathur, S.; Alexander, D.C.; Berger, A.; Shuster, J.R.;  Wei, J.T.; Varambally, S.; Beecher. C.; Chinnaiyan, A.M. Metabolomic profiles delineate potential role for sarcosine in prostate cancer progression. Nature, 2009, 457, 910–914.
  • Qiu,Y.P.; Cai, G.X.; Zhou, B.S.; Li, D.; Zhao, A.H.; Xie, G.X.; Li, H.K.; Cai, S.J.; Xie, D.; Huang, C.Z.; Ge, W.T.; Zhou, Z.X.; Xu, L.; Jia, W.P.; Zheng, S.; Yen, Y.; Jia, A Distinct Metabolic Signature of Human Colorectal Cancer with Prognostic Potential. Clinical Cancer Research, 2014, 20(8), 2136-2146.

Comments 3: L19-20....might provide useful insights into the correlations between the differential compounds and the quality of agarwood against anxiety disorder.>>> the sentence is vague please improve and write a proper CONCLUSION!  may have correlations components and the quality of agarwood preparations...

Response: We thank the reviewer for this correction. We have revised the abstract part on Line 17. “Brunei and Nha Trang agarwood had abundant sesquiterpenoids such as agarospirol, nootkatone, α-eudesmol, exhibiting notable pharmacological efficacy in relieving anxiety. Malaysia and Irian agarwood had abundant alcohols and aldehydes, qualifying them as high-quality spices. A  “compound-target-disease” network and pathway enrichment analysis were further employed to predict 79 gene targets and 20 pathways associated with the anxiolytic effects based on the 62 sesquiterpenoids. Correlated relationships among sesquiterpenoids and targets suggest that agarwood treats anxiety with multiple compounds acting on multiple targets. Varying levels of sesquiterpenoids across agarwood groups might lead to differences in anxiolytic effect via signaling pathways including neurotransmitter- and hormone-regulated pathways. Our study originally evaluate agarwood quality and its anxiolytic effect by linking characteristic compounds to potential gene targets and pathways.

  1. introduction

Comments 4: L25 (Aquilaria malaccensis)>>> (Aquilaria malaccensis Lam.) please add the scientific name to the mans.

Response: Thanks. We have changed the name on Line 30.

Comments 5: L30-32 have a format problem.. please allign correctly.

Response: Thanks. We have made corresponding changes on Line 30.

Comments 6: L57 .. To address these knowledge gaps, our work>>> To address the knowledge gaps, the present work..

Response: Thanks. We have made corresponding changes on Line 62.

Comments 7: in the results part, the authors should highlight their orginal findings? what was discovered for the first time?

Response: This is a good point. An original finding of our study is that varying levels of sesquiterpenes across agarwood groups would lead to differences in some signaling pathways such as neurotransmitter-related pathways, hormone-regulated pathways, and signal transduction pathways. To the best of our knowledge, this is the first study to evaluate agarwood quality and its anxiolytic effect by linking characteristic compounds to potential gene targets and pathways. We have highlighted this finding on Line 22 and Line 357.

Comments 8: The mans needs major revisions before it can be considered for publication

Response:Thanks. We agree with the reviewer’s comments. We have revised more related publications in the abstract and introduction section in the revised manuscript.  We further highlighted our findings in the Results and Discussion.

Round 2

Reviewer 2 Report

Comments and Suggestions for Authors

The authors addressed most of my comments and suggestions.

I wish they could improve the work much more, as the topic is quite interesting.

The conclusion still seems like an abstract of the work.. It should  have just CONCLUSION - suggestions.. no need to repeat the works details!

If the editorial accepts the work and format, I have no longer an objection. 

Author Response

Thank you very much for taking the time to review this manuscript.

We have revised the the Conclusions according to your suggestion. Also, we have check that all references are relevant to the contents of the manuscript. We  have revised and added some sentences and references. We have further highlighted the revisions to the manuscript.

Our point-by-point answers are provided  in the attachment.
